# Potential Anti-Alzheimer Properties of Mogrosides in Vitamin B12-Deficient *Caenorhabditis elegans*

**DOI:** 10.3390/molecules28041826

**Published:** 2023-02-15

**Authors:** Denia Cai Shi, Chunlin Long, Ella Vardeman, Edward J. Kennelly, Michael A. Lawton, Rong Di

**Affiliations:** 1New Brunswick Graduate School, Rutgers University, New Brunswick, NJ 08901, USA; 2College of Life and Environmental Sciences, Minzu University of China, Beijing 100086, China; 3Graduate Center, City University of New York, 365 5th Ave, New York, NY 10016, USA; 4Department of Biological Sciences, Lehman College, City University of New York, 250 Bedford Park Blvd West, Bronx, NY 10468, USA; 5Institute of Economic Botany, The New York Botanical Garden, 2900 Southern Boulevard, Bronx, NY 10458, USA; 6Department of Plant Biology and Pathology, School of Environmental and Biological Sciences, Rutgers University, New Brunswick, NJ 08901, USA

**Keywords:** mogrosides, Alzheimer’s disease, vitamin B12 deficiency, *Caenorhabditis elegans*

## Abstract

Vitamin B12 deficiency can lead to oxidative stress, which is known to be involved in neurodegenerative diseases such as Alzheimer’s disease (AD). Mogrosides are plant-derived triterpene glycosides that exhibit anti-inflammatory and antioxidant activity in animal cell lines and mouse models. Since amyloid-β toxicity is known to cause oxidative stress and damage to brain cells, we hypothesized that mogrosides may have a protective effect against AD. In this study, we investigated the potential anti-AD effect of mogrosides in vitamin B12-deficient wild-type N2 and in transgenic CL2355 *Caenorhabditis elegans* expressing amyloid-β peptide. Our data indicated that mogrosides have a beneficial effect on the lifespan and egg-laying rate of N2 and vitamin B12-deficient N2 worms. Additionally, the results revealed that mogrosides can effectively delay the paralysis of CL2355 worms as determined by serotonin sensitivity assay. Our analysis showed that mogrosides increase the expression of oxidative protective genes in N2 worms fed with vitamin B12-deficient OP50 bacterium. We conclude that mogrosides may exert preventative rather than curative effects that counteract the detrimental vitamin B12-deficient environment in N2 and CL2355 *C. elegans* by modulating oxidation-related gene expression.

## 1. Introduction

Current Alzheimer’s disease (AD) medications are not particularly effective [1] and have unfavorable side effects [2]. There is a pressing need for treatments and medications that can prevent the onset of AD or ameliorate and even reverse its pathology. Vitamin B12 deficiency has been associated with various diseases, including AD and diabetes [3,4]. Vitamin B12, an important micronutrient, is required for optimal neurocognitive functions [5] and its deficiency can lead to oxidative stress [6]. Recent reports have shown that plant-derived mogrosides possess not only anti-diabetic and antioxidant activity in vitro [7], but also lower lipid levels, and reduce oxidative stress and serum glucose levels in diabetic mice [8]. Diabetes and AD share some risk factors, such as increased levels of amyloid-β (Aβ) deposition, together with elevated cholesterol and increased oxidative stress [9]. Another link between diabetes and AD is that changes in glycemic control are correlated with cognitive performance [10]. These prompted us to investigate the potential of antioxidant and anti-inflammatory mogrosides, which are triterpene glycosides produced by *Momordica grosvenori* [11]. In this study, we assayed the ability of mogrosides to ameliorate or counter the effects of vitamin B12 deficiency on wild type N2 *C. elegans* and on transgenic CL2355 *C. elegans* that express amyloid-β peptide [12,13].

There has been renewed interest in exploiting plants used in traditional medicine to identify and understand the mode-of-action of specific active components responsible for medical efficacy [14]. Mogrosides are a mixture of triterpene glycosides found in *M. grosvenori* (Luo Han Guo, LHG), which is native to China and has been used as an anti-inflammatory herbal medicine for thousands of years [11]. The Food and Drug Administration (FDA) has granted this compound the status of Generally Recognized as Safe (GRAS) to be used as a non-caloric natural sweetener. Mogrosides are known to have important biological activities [15]. Previous research has shown that mogrosides exert an anti-inflammatory effect by up-regulating the expression of protective inflammatory genes such as *BCL2l1*, *PARP1* and *TRP53* and by down-regulating the expression of inflammatory genes such as *COX-2*, *IL-6* and *iNOS* in murine macrophage RAW 264.7 cells [11]. Additionally, recent reports have shown that mogrosides possess antioxidant activity in vitro [7], which may help them exert a neuroprotective effect [16].

The nematode worm *C. elegans* has been used as a model system for studying various diseases and human pathologies. About 83% of the *C. elegans* proteome has a homologous human sequence [17]. Although WT N2 *C. elegans* do not naturally produce the Aβ peptide associated with AD, transgenic *C. elegans* expressing the human Aβ peptide have allowed key aspects of the disease to be modeled and studied in this system [18]. Several natural compounds, such as benzylphenethylamine alkaloids from *Lycoris* spp., coffee extracts and resveratrol [2,19,20] have been tested for anti-AD properties using *C. elegans*. In CL2355 worms, a shift in temperature from 16 °C to 22 °C induces gene expression and the production of human Aβ peptide within neuronal cells [21]. Significantly, these neuronal cells accumulate insoluble Aβ, a condition also observed in AD patients [13]. This provides an experimental platform for understanding the genetic and environmental factors that contribute to the formation of insoluble Aβ and also provides an assay for assessing treatments that might ameliorate this pathology. We have recently used *C. elegans* CL2355 worms to investigate the anti-AD effects of alkaloid extracts from various *Lycoris* spp. [22]. Previous studies have highlighted a link between vitamin B12 deficiency and AD [23]. Consequently, in this study, we explored the effects of mogrosides on N2 and CL2355 *C. elegans* under regular and vitamin B12-deficient conditions.

Both neuroinflammation and Aβ deposition are associated with the progression of AD. Since neuroinflammation involves the secretion of pro-inflammatory cytokines [24] and Aβ toxicity is known to cause oxidative stress and damage to brain cells [13], we were interested in whether mogrosides might exert a protective effect against AD by preventing or alleviating damage resulting from Aβ deposition, neuroinflammation, and oxidative stress. Consequently, we exposed both N2 and CL2355 worms to mogrosides under regular and vitamin B12-deficient conditions and examined the expression of genes associated with inflammation, oxidative stress, neuroprotection and the onset of AD in humans. This study aims to demonstrate the potential anti-AD properties of mogrosides by analyzing how mogrosides modulate the expression of genes that are related to vitamin B12 deficiency and AD.

## 2. Results

### 2.1. Chemical Composition

The 90% pure mogroside extract was a gift from Ibis America, LLC (Branchburg, NJ, USA). The mogrosides were extracted and quantified by HPLC as previously reported [25]. The level of mogroside V was determined to be 25.9% in the mixture. The composition of the extracts of mogrosides was re-determined by the UPLC-qToF-MS method (Table 1 and Appendix A). The analytical results (Figure 1A–E) showed that mogroside V was the most abundant compound in our mogroside extract.

### 2.2. Effects of Mogrosides on N2 C. elegans Lifespan

Lifespan analysis was carried out over a period of 19 days on L4 stage N2 and N2/B12^−^ (B12 deficient) worms grown on NGM agar plates in the presence of mogrosides and fed with either OP50 or OP50/B12^−^. Changes in lifespan were measured by determining the day at which 50% viability was reached, calculated using a polynomial regression line for each of the curves. The results (Figure 2A) show that the lifespan of N2 + OP50/B12^−^ was reduced to 12 days compared to N2 + OP50 where lifespan was 12.7 days (*p* = 0.2). Mogrosides extended the lifespan of N2 + OP50 from 12.7 days to 15.8 days (*p* = 0.6). However, mogrosides did not seem to improve the lifespan of N2 + OP50/B12^−^, which showed a slightly reduced 50% viability of 11.8 days (*p* = 0.4) compared to the 12 days observed without mogroside treatment.

In separate experiments, we found that the lifespan of B12-deficient N2 (N2/B12^−^) + OP50/B12^−^ was markedly reduced to 10.9 days, compared to N2/B12^−^ + OP50, which had a lifespan of 14.2 days (*p* = 0.01) (Figure 2B). Mogrosides had a significant effect on lifespan of N2/B12^−^ worms fed with either OP50 or OP50/B12^−^, extending it from 10.9 days, in the absence of mogrosides, to 15.5 days (*p* = 0.01) and 13 days (*p* = 0.1), respectively, in the presence of mogrosides. Together, these results indicate that mogrosides can help extend lifespan and, in some circumstances, counter the deleterious effects of vitamin B12-deficiency.

### 2.3. Effects of Mogrosides on C. elegans Reproduction

Since vitamin B12 deficiency is known to reduce the fertility of N2 worms [12], we evaluated the effects of mogrosides on egg-laying capacity of N2 and N2/B12^−^. The average number of eggs laid by N2 and N2/B12^−^ during the first 3 days after continued exposure to mogrosides under regular and vitamin B12-deficient conditions is presented in Figure 3. N2 + OP50/B12^−^ exhibited a 33.01% reduction in the number of eggs produced on the first day of exposure, compared to N2 + OP50 (*p* = 0.109) (Figure 3A). Furthermore, N2 + OP50/B12^−^ produced 42.32% and 35.84% fewer eggs than did N2 + OP50 on the second (*p* = 0.080) and the third days (*p* = 0.154), respectively. Mogrosides did not significantly increase the number of eggs produced by N2 + OP50 during the first and second days of exposure. However, on the third day, N2 + OP50 + mogrosides produced 41.52% more eggs than N2 + OP50 (*p* = 0.199). Similarly, N2 + OP50/B12^−^ + mogrosides exhibited an increase in eggs laid by 30.80% on the second (*p* = 0.229) day and 42.23% on the third (*p* = 0.200) day, compared to N2 + OP50/B12^−^.

Mogrosides increased the number of eggs laid by N2/B12^−^ + OP50 by 27.89% (day 2, *p* = 0.111) and 40.14% (day 3, *p* = 0.111), compared to N2/B12^−^ + OP50 without mogroside treatment, respectively (Figure 3B). Similarly, mogrosides also increased eggs laid by N2/B12^−^ + OP50/B12^−^ by 36.43% (day 2, *p* = 0.116) and 58.05% (day 3, *p* = 0.103), compared to N2/B12^−^ + OP50/B12^−^ without mogroside treatment. These results are in concert with those found for lifespan and suggest that mogrosides also have a net beneficial effect on worm fecundity under both normal and vitamin B12-deficient conditions.

### 2.4. Effects of Mogrosides on the Expression of Genes Involved in Inflammation, EGF and Oxidation

We investigated the effects of mogrosides on the expression of *C. elegans* genes involved in inflammation, neuroprotection and oxidation. Oxidation-related genes that are involved in protection against harmful reactive oxygen species (ROS) include *sod-1*, *gpx-1*, *ctl-1* and *gst-10* [26]. Epidermal growth factor (EGF) genes associated with neuroprotection include *epi-1*, *lrp-1* and *lrx-1* [27]. Inflammation-related genes involved in the innate immune response include *trf-1*, *ZC239.12* and *F22E5.6* [28]. Two *C. elegans* genes encoding heat-shock proteins with functions similar to human *hsp-16.2* and *hsp-16.41* were included as reporters of heat shock stress and the unfolded protein response [2]. The known functions of these *C. elegans* genes and their respective human orthologs are summarized in Table 2.

#### 2.4.1. Effects of Mogrosides on the Expression of Oxidation-Related Genes in N2, N2/B12^−^, CL2355 and CL2355/B12^−^
*C. elegans*

RT-qPCR was used to evaluate the effect of mogrosides on the expression of *sod-1*, *gpx-1*, *ctl-1* and *gst-10* genes in N2, N2/B12^−^, CL2355 and CL2355/B12^−^ *C. elegans*. Our data show that 2 mM mogrosides slightly increased the expression of *gpx-1* (Figure 4B), *clt-1* (Figure 4C) and *gst-10* (Figure 4D) in N2 + OP50 compared to non-treated N2 + OP50. The data also show that the expression of *sod-1* (Figure 4A), *gpx-1* (Figure 4B) and *clt-1* (Figure 4C) was increased slightly in N2 fed with OP50/B12^−^. However, 2 mM mogrosides further significantly enhanced expression of these three genes in these worms by 1.25-fold (*p* = 0.018) (Figure 4A), *gpx-1* by 2.75-fold (*p* = 0.0003, Figure 4B) and *ctl-1* by 1.6-fold (*p* = 1.65751 × 10^−5^ Figure 4C), respectively, compared to N2 + OP50. In contrast, mogrosides did not appear to have the same effect on the expression of these three genes in B12-deficient worms (N2/B12^−^ + OP50 and N2/B12^−^ + OP50/B12^−^) (Figure 4A–C). The expression of *gst-10* was not up-regulated in any of the worms, whether they were fed with OP50/B12^−^ or treated with mogrosides (Figure 4D).

For CL2355 and CL2355/B12^−^ worms fed with OP50 or OP50/B12^−^, the expression of *sod-1*, *gpx-1* and *ctl-1* genes was up-regulated 2- to 5.5-fold in the absence of mogrosides (Figure 5A–C), compared to N2 + OP50. The addition of 2 mM mogrosides did not further enhance the expression of these three genes. The expression of *gst-10* increased three-fold (*p* = 0.002) in CL2355 + OP50/B12^−^ and by a factor of 4.7 (*p* = 0.001) in CL2355/B12^−^ + OP50/B12^−^, compared to N2 + OP50 (Figure 5D). Mogrosides increased the expression of *gst-10* in CL2355 + OP50/B12^−^ and CL2355/B12^−^ + OP50/B12^−^ by factors of 2.7 (*p* = 0.0002) and 3.6 (*p* = 0.01), respectively, compared to N2 worms + OP50 (Figure 5D).

#### 2.4.2. Antioxidant Activity of Mogrosides on N2, N2/B12^−^, CL2355 and CL2355/B12^−^ *C. elegans*

Oxidative stress has been widely associated with both Aβ toxicity and the pathology of AD [33]. DCF-DA dye was used to measure intracellular levels of H_2_O_2_-related ROS in N2, N2/B12^−^, CL2355 and CL2355/B12^−^ worms treated with 2 mM mogrosides under regular and vitamin B12-deficient conditions.

Our data show that 2 mM mogrosides reduced H_2_O_2_ levels by 19% (*p* = 5.626 × 10^−6^) in N2 + OP50 (Figure 6A). In N2 + OP50/B12^−^, the H_2_O_2_ level increased by 6% (*p* = 3.545 × 10^−5^) and mogrosides reduced it by 28% (*p* = 1.960 × 10^−5^), compared to N2 + OP50 (Figure 6A). In N2/B12^−^ + OP50, the H_2_O_2_ level was slightly reduced (9%, *p* = 8.994 × 10^−5^), and the application of 2 mM mogrosides did not reduce it further (Figure 6A), compared to the level in N2 + OP50. In contrast, in N2/B12^−^ + OP50/B12^−^, H_2_O_2_ levels decreased by 23% (*p* = 7.947 × 10^−3^), while treatment with 2 mM mogrosides led to an even more marked reduction, with H_2_O_2_ levels falling by 56% (*p* = 2.373 × 10^−5^), compared to N2 + OP50 (Figure 6A). For CL2355 worms, there was a general decrease in H_2_O_2_ levels, whether or not they were fed with OP50 or OP50/B12^−^ (Figure 6B). The addition of 2 mM mogrosides further reduced H_2_O_2_ levels in CL2355 + OP50/B12^−^ by 50% (*p* = 2.914 × 10^−5^) (Figure 6B).

#### 2.4.3. Effects of Mogrosides on EGF Genes in N2, N2/B12^−^, CL2355 and CL2355/B12^−^ *C. elegans*

We evaluated the ability of mogrosides to modulate gene expression for *lrp-1*, the *C. elegans* homolog of mammalian epidermal growth factor (EGF)-related genes, known to be involved in neuroprotection and the transport and endocytosis of the Aβ complex in mammals [27]. Our results indicate that the expression of *lrp-1* did not change in N2 + OP50/B12^−^, while 2 mM mogrosides increased the expression of *lrp-1* by factors of 1.15 and 1 in N2 + OP50 and N2 + OP50/B12^−^, respectively, compared to non-mogrosides-fed N2 + OP50 (*p* = 0.006) (Figure 7). Expression of *lrp-1* significantly decreased (*p =* 0.006) in N2/B12^−^ + OP50, compared to N2 + OP50, while 2 mM mogrosides up-regulated *lrp-1* in N2/B12^−^ + OP50, but not in N2/B12^−^ + OP50/B12^−^ (*p* = 0.027) (Figure 7).

The expression of *lrp-1* in CL2355 + OP50 increased (*p* = 0.052) by a factor of 3.8, compared to N2 + OP50, which may reflect a response to expression of human Aβ peptide following induction of its expression (Figure 8). Furethermore, 2 mM mogrosides very significantly (*p* = 0.001) up-regulated *lrp-1* expression further by a factor of 6.2 (Figure 8). When CL2355 worms were fed with OP50/B12^−^, *lrp-1* increased by a factor of 6.8 (*p* = 0.012), while 2 mM mogrosides up-regulated its expression by a factor of 4.6 (*p* = 0.02) (Figure 8). In CL2355/B12^−^ + OP50, *lrp-1* expression increased five-fold (*p* = 0.014) while 2 mM mogrosides up-regulated its expression by a factor of 3.7 (*p* = 0.037) (Figure 8). The CL2355/B12^−^ + OP50/B12^−^ showed a three-fold increase in *lrp-1* expression (*p* = 0.017), while 2 mM mogrosides only up-regulated its expression by a factor of 0.6 (*p* = 0.186) (Figure 8).

The expression of *epi-1* and *lrx-1* genes was investigated for their possible neuroprotective role in worms. Our data showed that the expression of *epi-1* and *lrx-1* genes did not change in N2, N2/B12^−^, CL2355 and CL2355/B12^−^ worms and that mogrosides treatment had no effect on their expression in either N2 or CL2355 worms.

#### 2.4.4. Effects of Mogrosides on the Expression of Inflammation and Stress Response Genes in N2, N2/B12^−^, CL2355 and CL2355/B12^−^ *C. elegans*

In addition to antioxidant and EGF genes, inflammatory and stress responses have also been associated with anti-AD mechanisms [24]. Consequently, we investigated the expression of *trf-1* (homolog of *TRAF3*, a tumor necrosis factor involved in innate immune response), *ZC239.12* and *F22E5.6* (homolog of *TNFA1P1*, a tumor necrosis factor involved in the innate immune response), *hsp-16.2* and *hsp-16.41* (homologs of *hsp* genes involved in stress response) in N2, N2/B12^−^, CL2355 and CL2355/B12^−^ worms. These studies showed that the expression of these genes was unchanged in response to mogrosides treatment, compared to N2 + OP50 and CL2355 + OP50.

In summary, RT-qPCR analysis shows that mogroside treatment induces expression of genes involved in oxidative stress, while (with the exception of *lrp-1*) other genes surveyed that are involved in the inflammatory response, neuroprotection and the heat-shock response are not induced by this treatment.

### 2.5. Serotonin Sensitivity Assay of CL2355 and CL2355/B12^−^ C. elegans by Mogrosides

Serotonin sensitivity assays revealed that treatment with 2 mM mogrosides affected the time it took for CL2355 + OP50 to become paralyzed, increasing it from 19 to 38 min (*p* = 0.0002) (Figure 9). Application of 2 mM mogrosides increased the time it took for CL2355/B12^−^ + OP50/B12^−^ to become paralyzed from 17 to 29 min (*p* = 4.58087 × 10^−6^). For CL2355/B12^−^ + OP50, treatment with 2 mM mogrosides increased time-to-paralysis from 17 to 37 min, compared to CL2355 + OP50 (*p* = 3.23858 × 10^−8^). These results further support the conclusion that mogrosides have a beneficial effect on neuromuscular performance and the overall well-being of worms.

### 2.6. Effect of Mogrosides on the Acetylcholine Esterase Gene Expression in Aβ-Transgenic CL2355 and CL2355/B12^−^ C. elegans

In humans, acetylcholine esterase (*AChE*) is involved in the breakdown of the neurotransmitter acetylcholine, and functions to terminate synaptic transmission between neurons [34]. We have previously shown that alkaloids from *Lycoris* spp. can inhibit the gene expression of both *ace-1* and *ace-2* and rescue CL4176 worms from paralysis caused by expression of Aβ in muscle cells [2]. In *C. elegans*, it is known that 95% of the enzymatic activity of *AChE* comes from both *ace-1* and *ace-2* [35]. Here, we focused on *ace-2* which is expressed only in neurons [2], whereas *ace-1* is expressed solely in the vulva and in body muscle cells. Significantly, we recently showed that application of *Lycoris* alkaloids inhibits *ace-2* expression in CL2355 worms [22].

To investigate the anti-paralysis mechanisms of mogrosides in CL2355 and CL2355/B12^−^ worms, we examined the ability of mogrosides to inhibit *ace-2* gene expression in CL2355 and CL2355/B12^−^ worms. RT-qPCR analysis showed that 2 mM mogrosides did not alter *ace-2* expression in CL2355 + OP50 (Figure 10. When CL2355 worms were fed with OP50/B12^−^, *ace-2* expression was reduced by 25% (*p* = 0.056), compared to CL2355 + OP50. However, 2 mM mogrosides treatment did not further alter the expression of *ace-2* in CL2355 + OP50/B12^−^ (Figure 10). Similarly, 2 mM mogrosides did not affect *ace-2* expression in CL2355/B12^−^ + OP50, or in CL2355/B12^−^ + OP50/B12^−^, which displayed a 40% reduction in *ace-2* expression, regardless of mogrosides treatment (*p* = 0.002) (Figure 10). These results indicate that the reduction of *ace-2* expression in CL2355 resulted from feeding these worms with OP50/B12^−^ and was not affected by treatment with 2 mM mogrosides.

## 3. Discussion

Prior studies have noted the anti-inflammatory and antioxidant activity of mogrosides [8,11], suggesting that mogrosides may have protective effects against neurodegenerative diseases [36]. AD has been steadily increasing worldwide and current medications are not particularly effective [1] and have unfavorable side effects [2]. In this study, we used vitamin B12-deficient N2 mutant and Aβ-expressing CL2355 *C. elegans* to investigate the potential anti-AD properties and mode-of-action of natural product mogrosides. Specifically, we examined the effects of these compounds on lifespan, reproduction, gene expression and serotonin sensitivity. Our results demonstrate that mogrosides extend lifespan of both N2 and N2/B12^−^ worms under both regular and vitamin B12-deficient conditions (Figure 2). Additionally, application of 2 mM mogrosides increased the number of eggs laid by N2 and N2/B12^−^ worms after 2 and 3 days, compared to N2 and N2/B12^−^ worms fed with only OP50 or OP50/B12^−^ (Figure 3), indicating that mogrosides have the potential to restore fertility in vitamin B12-deficient worms.

Our mogrosides extract was composed of mogrosides IIe, III, IV and V, where the mogroside V was the most abundant in the mixture (Figure 1). Mogroside V is known to have protective effects against oxidative stress in skin fibroblast [37] and neuronal damages in mice [38], which supports our gene expression results. Gene expression studies using RT-qPCR indicated that mogrosides up-regulated the expression of oxidative-related genes, including *sod-1* (Figure 4A), *gpx-1* (Figure 4B) and *clt-1* (Figure 4C) in N2 fed with OP50/B12^−^, which supports previous research findings regarding the antioxidant properties of mogrosides [7]. In CL2355 and CL2355/B12^−^ worms, fed with either OP50 or OP50/B12^−^, expression of the oxidative genes *sod-1* (Figure 5A), *gpx-1* (Figure 5B), *clt-1* (Figure 5C) and *gst-1* (Figure 5D) were increased, suggesting a possible coping response to oxidative stress conditions in CL2355 worms that result from the ectopic expression of Aβ peptides. Furthermore, 2 mM mogrosides did not further elevate the expression of these four genes in CL2355 and CL2355/B12^−^ worms (Figure 5A–D).

The antioxidative properties of mogrosides were further supported by direct measurement of H_2_O_2_ levels, which showed that treatment with 2 mM mogrosides was highly effective in reducing ROS levels in N2 + OP50, N2 + OP50/B12^−^, N2/B12^−^ + OP50 and N2/ B12^−^ + OP50/B12^−^ (Figure 6A). Interestingly, mogrosides were less effective in reducing H_2_O_2_ levels in CL2355 (Figure 6B), compared to strains not expressing Aβ-peptide. This observation correlates with the effects of mogrosides on the expression of anti-oxidative genes, at least in N2 and N2/B12^−^ worms.

We have previously shown that mogrosides display anti-inflammatory activity in macrophage RAW 264.7 cells and in a mouse ear model [1]. Here, we investigated the expression of inflammation- and neuroprotection-related genes including *epi-1*, *lrp-1*, *lrx-1*, *trf-1*, *ZC239.12*, *F22E5.6*, *hsp-16.2* and *hsp-16.41* in N2 and CL2355 worms fed with OP50 or OP50/B12^−^ in the presence or absence of 2 mM mogrosides. Our data show that with the exception of *lrp-1*, the expression of these genes was not significantly affected by mogrosides. However, mogrosides did increase the expression of *lrp-1* by a factor of 1.15 in N2 + OP50 and by a factor of 0.9 in N2 + OP50/B12^−^ (Figure 7). Mogrosides also increased *lrp-1* expression level by a factor of 6.2 in CL2355 + OP50 (Figure 8). However, mogrosides did not further increase *lrp-1* expression in B12-deficient N2 or CL2355 worms, whether or not they were fed with OP50 or OP50/B12^−^ (Figure 7). *C. elegans lrp-1* closely resembles human *LRP2*, which is known to be involved in the transport and endocytosis of the Aβ complex [31] and has been implicated in Aβ clearance [18]. The decrease in expression of *lrp-1* has been shown to affect neurotransmission in *C. elegans* [18]. Our results suggest that mogrosides may exert a preventative rather than curative effect on neurons under vitamin B12 deficiency conditions.

Temperature-shifted CL2355 worms express Aβ-peptide in neural cells, resulting in paralysis, as displayed both phenotypically and by the serotonin sensitivity assay [13]. Our experiments showed that 2 mM mogrosides delayed the onset of paralysis in CL2355 + OP50. We further observed that mogrosides extended the paralysis time of CL2355/B12^−^ fed with either OP50 or OP50/B12^−^ bacteria, suggesting that mogrosides may have an inhibitory effect against Aβ toxicity. However, our RT-qPCR results show that *ace-2* gene expression decreased in CL2355 and CL2355/B12^−^ worms fed with OP50/B12^−^, and that 2 mM mogrosides did not further modulate *ace-2* expression levels. The human *AChE* homolog is known to be involved in the pathogenesis of AD [39]. However, our data suggest that the anti-paralysis properties of mogrosides in CL2355 do not arise from its modulation of *ace-2* gene expression.

Overall, our results demonstrate that mogrosides significantly increase lifespan and reproduction rate in both N2 and N2/B12^−^ worms under regular and vitamin B12 deficiency conditions. In CL2355 and CL2355/B12^−^ worms, mogrosides treatment delayed the onset of paralysis, although the underlying cellular mechanisms responsible for this beneficial effect remain to be determined. It would be particularly interesting in future studies to examine the levels and sub-cellular distribution of Aβ peptides in neuronal cells. This might help determine whether the beneficial effects of mogrosides are the result of reduced accumulation of insoluble Aβ peptides or, rather, represent a separate, ameliorative pathway. To this point, our data suggest that the up-regulation of oxidation protective genes may significantly contribute to the ability of mogrosides’ to protect worms from the deleterious effects of Aβ expression and accumulation. In addition, modulation of *lrp-1* gene expression by mogrosides may also contribute to the protective effects of mogrosides in N2 and CL2355 worms.

## 4. Materials and Methods

### 4.1. Mogrosides Extraction and Analysis by UPLC-qToF-MS Analysis

Mogrosides extract was isolated from LHG and obtained from Ibis America, LLC (Branchburg, NJ, USA) as described [25]. The extraction was then re-analyzed by UPLC-qToF-MS to confirm its purity and mogroside contents. An ACQUITY ultrahigh-performance liquid chromatograph (UPLC) paired with a Xevo G2 QToF-MS (Waters Corporation, Milford, MA, USA) was used to identify and compare the relative abundance of mogrosides. Powdered extract was prepared at 2 mg/mL concentration in (*v*/*v*) 80% MS-grade methanol: water and filtered using Whatman (Maidstone, United Kingdom) Uniflo 0.45 µm syringe filters. The UPLC used an ACUITY UPLC BEH C18 1.7 μm, 2.1 × 50 mm column (Waters Corporation, Milford, MA, USA). The solvent system consisted of MS-grade water with 0.1% formic acid (solvent A) and MS-grade acetonitrile with 0.1% formic acid (solvent B). The UPLC method was adapted from a previously published method [40]. Using a 0.25 mL/min flow rate, the gradient was as follows: 0–8.00 min, 15–30% B; 8.00–8.50 min, 30–85% B; 8.50–9.50 min, 85% B; 9.50–10 min, 85–15% B; 10.00–12.00 min, 15% B. The column was maintained at 40 °C. The extract was analyzed in triplicate with 2 µL injections with needle overfill. Lock mass correction was performed with a solution of leucine–enkephalin. Electrospray ionization was used in negative polarity mode (ESI–). The settings were as follows: sample probe capillary voltage, 2.5 kV; sampling cone voltage, 50 V; extraction cone voltage, 4 V; source temperature, 100 °C; and desolvation temperature, 250 °C. Nitrogen was used as the desolvation flow rate of 800 L/h. Data were acquired using MS^e^ over a mass range of 100–1500 Da with a scan time of 0.2 s.

### 4.2. Preparation of Vitamin B12-Deficient OP50 Bacterium

B12-deficient OP50 was prepared as described by Bito et al., 2013. OP50 was inoculated into a 50 mL falcon tube containing 100% LB media and grown at 37 °C for 1 day. Then, 100 μL of the 100% LB media cell culture was transferred to a new 50 mL falcon tube containing 50% LB and 50% M9 liquid media and grown at 37 °C for 1 day. For the next three days, 100 μL of the cell culture was transferred to a new LB-M9 solution with half the percentage of LB used on the previous day. On the fourth day, 100 μL of the cell culture was transferred to a 100% M9 solution. OP50/B12^−^ was subsequently maintained on 100% M9 solution.

### 4.3. Acquisition and Preparation of Vitamin B12-Deficient N2 and CL2355 Worms

N2 and CL2355 *C. elegans* were obtained from the Caenorhabditis Genetics Center (CGC). Vitamin B12-deficient N2 worms were prepared as described previously [12]. One adult N2 worm was transferred to an M9 medium supplemented with cyanocobalamin and plated with OP50/B12^−^. After the adult N2 worm had laid eggs, it was picked from the plate. Eggs were left to grow for 3 days until they had reached the full adult stage. The eggs laid by these adult worms were picked and transferred to new M9 medium containing cyanocobalamin with OP50/B12^−^. This step was repeated for 10 generations. After the 10th generation, the eggs were picked to an M9 plate without cyanocobalamin for 5 generations. Subsequently, N2/B12^−^ worms were maintained on M9 medium without cyanocobalamin. All procedures involving N2 worms were carried out at room temperature (22 °C). The same procedures were repeated for vitamin B12-deficient CL2355 worms, except that the time CL2355 worms took to reach the full adult stage was 5–6 days and grown at 16 °C, the temperature that is non-permissive for the expression of the human Aβ peptide.

### 4.4. Lifespan and Reproduction Assay

Nematode Growth Media (NGM) was prepared in 60 mm petri dishes to maintain *C. elegans*. *E. coli* strain OP50 was used as a food source for *C. elegans*. M9 medium with agar was prepared as shown previously [12] in 35 mm petri dishes for treatments. Mogrosides at 2 mM were mixed with either OP50 or OP50/B12^−^ and incorporated into an M9 medium in 35 mm petri plates for the lifespan analysis. N2 and N2/B12^−^ *C. elegans* were synchronized by harvesting the eggs from the adult worms using the bleach method (www.wormbook.org) (accessed on 24 December 2022). The eggs were washed twice with M9 buffer and then once with S buffer. Isolated N2 eggs were plated on NGM with only OP50 until the worms reached the L4 stage. N2/B12^−^ isolated eggs were plated on an M9 medium with only OP50/B12^−^ until the worms reached the L4 stage. Then, they were transferred to M9 medium plates containing mogrosides and either OP50 or OP50/B12^−^.

Approximately 30 synchronized N2 and N2/B12^−^ worms at the L4 stage were plated into treatment-containing 35 mm plates. Worms were observed over a period of 19 days to record their viability. This process was repeated three times and the results were presented in a line graph with standard deviations. To calculate the number of eggs laid by the worms, the 35 mm plates were divided into equal quadrants. The number of eggs were counted on one quadrant and then multiplied by 4 to determine the number of eggs laid from day 1 to day 3 during the lifespan analysis. The total number of eggs was divided by the total amount of worms on the plate to obtain the number of eggs laid per worm per day.

### 4.5. RNA Isolation and RT-qPCR

N2 and N2/B12^−^ worms were harvested 3 days after treatment to isolate RNA for the RT-qPCR analysis. CL2355 and CL2355/B12^−^ worms were treated from the egg stage and grown at 16 °C, at the non-permissive state for the expression of the human Aβ peptide, until they reached the L4 stage (5 days). Then, the L4 worms were then shifted to 22 °C, where the conditions were permissive for inducing expression of the human Aβ peptide and collected after 3 days for RT-qPCR analysis. The worms were transferred into 1.5 mL Eppendorf tubes and Trizol was added. The freeze-cracking method was utilized to isolate RNA from the *C. elegans* in each treatment (www.wormbook.org) (accessed on 24 December 2022). The OD_260_ was measured using a Nano-spectrophotometer to determine the RNA concentration.

The reverse transcription reaction was conducted as specified in the Applied Biosystems High-Capacity cDNA Reverse Transcriptase Kit. The RT-qPCR primers were designed with PrimeExpress (Applied Biosystems) for all genes mentioned in Table 2. Primer sequences are listed in Table 3. SYBR Green master mix (Applied Biosystems) was used to conduct RT-qPCR with the respective primers to analyze the expression levels of the chosen genes. The actin gene was used as the endogenous control gene. The relative gene expression levels were calculated by the 2^−ΔΔCt^ method [41].

### 4.6. Serotonin Sensitivity Assay

CL2355 and CL2355/B12^−^ *C. elegans* were synchronized by harvesting eggs from adult worms using the bleach method. Eggs were washed twice with M9 buffer and then once with S buffer. Isolated eggs from CL2355 and CL2355/B12^−^ worms were plated on M9 medium containing mogrosides and either OP50 or OP50/B12^−^ and grown at 16 °C for 4–5 days (approximate late L3 stage). Then switched to 22 °C for 24 h until the worms reached the L4 stage. After they reached the L4 stage, worms were collected and washed with M9 buffer. Worms were then transferred to a 96-well plate for paralysis screening. Approximately 5 worms in 40 μL of M9 buffer were pipetted into each well. Each treatment consisted of 6 wells and about 30 worms in total. Serotonin at 10 mg/mL was prepared in M9 buffer and 40 μL was added to each well containing the worms (the final concentration of serotonin in each well was 5 mg/mL). The worms were then scored under the microscope for paralysis every 5 min until all worms were paralyzed.

### 4.7. H_2_O_2_ Assay for C. elegans

For the H_2_O_2_ assay, N2 and N2/B12^−^ worms were treated with 2 mM mogrosides from the egg stage and collected after 2 days at the L4 stage. CL2355 and CL2355/B12^−^ were also treated from the egg stage and collected at the L4 stage. Approximately 50 L4 worms from either treated N2, N2/B12^−^, CL2355 or CL2355/B12^−^ were collected into 100 µL 1× PBS and 1% Tween-20 in Eppendorf tubes. Worms were then sonicated at 30% amplitude for 15 s using a handheld sonicator and then chilled on ice for 45 s. This process was repeated 3 times. Afterward, the 100 µL sonicated worms were pipetted into 0.5 µL dichloroflourescin diacetate plus 99.5 µL 1× PBS in the wells of a 96-well plate. The plate was read in a 96-well plate reader (Agilent BioTek Synergy H1 Hybrid Multi-Mode Reader) at 37 °C with the excitation at 485 nm and the emission at 530 nm. These experiments were replicated threefold (biological replicas), and each data point resulted from three independent assays (technical replicas).

### 4.8. Statistical Analysis

The logrank test was used to access the statistical significance between the treatments in the lifespan analysis. We also conducted a Student *t*-test to assess the statistical significance between the non-treated and the variously treated N2 and CL2355 in RT-qPCR analysis, egg count analysis, serotonin sensitive assay and H_2_O_2_ assay. All data were presented as the mean ± SD. Differences were considered statistically significant when * *p* < 0.05, ** *p* < 0.01 and *** *p* < 0.001.

## Figures and Tables

**Figure 1 molecules-28-01826-f001:**
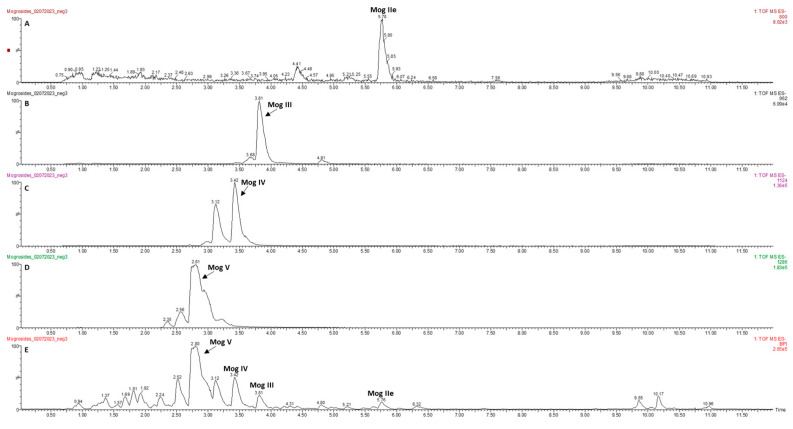
LC-MS base peak intensity extracted ion chromatograms at 800 *m*/*z* mogroside IIe (**A**), 962 *m*/*z* mogroside III (**B**), 1124 *m*/*z* mogroside IV (**C**), 1286 *m*/*z* mogroside V (**D**), total ion chromatogram (**E**).

**Figure 2 molecules-28-01826-f002:**
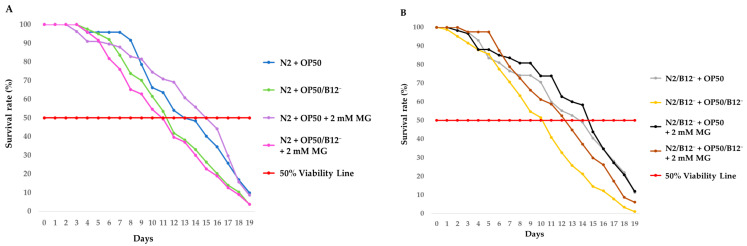
Lifespan of N2 and N2/B12^−^ worms. Synchronized N2 (**A**) and N2/B12^−^ (**B**) worms at the L4 stage were fed with either OP50 or OP50/B12^−^ with or without 2 mM mogrosides (MG). The percentage viability of N2 and N2/B12^−^ worms was plotted from day 0 to day 19. Three independent lifespan experiments were performed, and the results were used to calculate the average time at which 50% of worms survived. Statistical significance was compared to N2 worms + OP50 (**A**) or N2/B12^−^ worms + OP50/B12^−^ (**B**) and defined by the logrank test.

**Figure 3 molecules-28-01826-f003:**
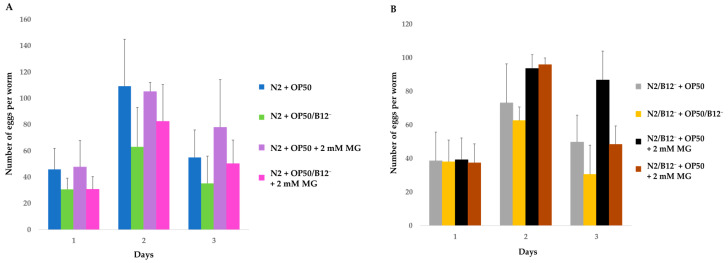
Average number of eggs laid by N2 (**A**) and N2/B12^−^ (**B**) worms in the first three days after the treatments indicated. Three independent assays were used to calculate the average number of eggs produced and the standard deviations. The Student *t*-test was conducted to assess the statistical significance.

**Figure 4 molecules-28-01826-f004:**
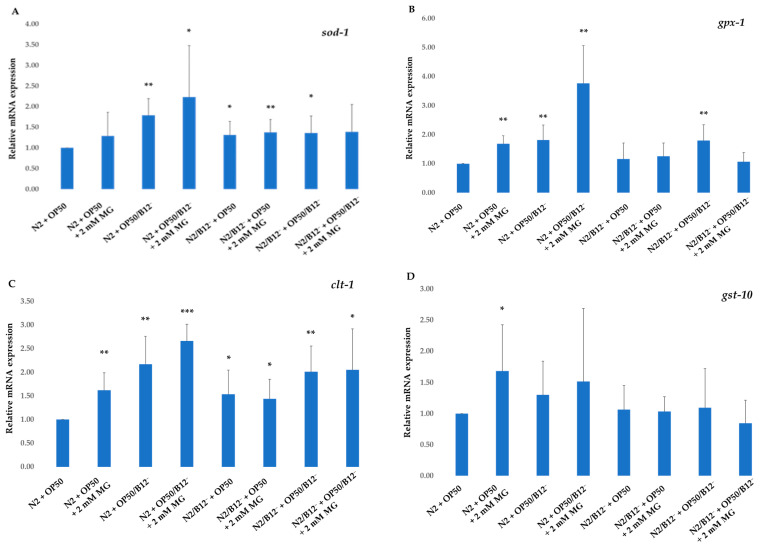
Expression of *sod-1* (**A**), *gpx-1* (**B**), *ctl-1* (**C**) and *gst-10* (**D**) genes on N2 and N2/B12^−^ fed with either OP50 or OP50/B12^−^ with or without 2 mM mogrosides. The fold-change (2^−ΔΔCt^) and standard deviations of the gene expression were calculated from five independent experiments. All treatments were compared to control N2 + OP50. The Student *t*-test was conducted to assess statistical significance (* *p* < 0.05; ** *p* < 0.01; *** *p* < 0.001).

**Figure 5 molecules-28-01826-f005:**
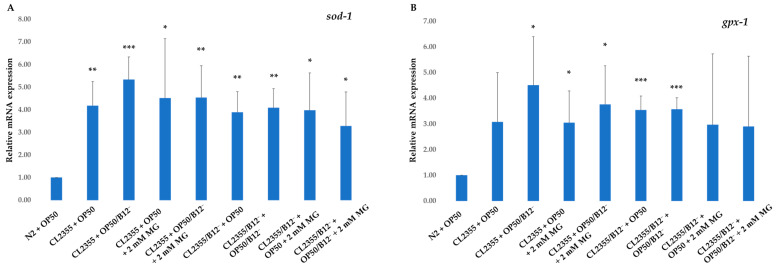
Expression of *sod-1* (**A**), *gpx-1* (**B**), *ctl-1* (**C**) and *gst-10* (**D**) genes in CL2355 and CL2355/B12^−^ worms fed with either OP50 or OP50/B12^−^ with or without 2 mM mogrosides (MG). The fold-change (2^−ΔΔCt^) and standard deviations of the gene expression were calculated from three independent experiments. All treatments were compared to N2 + OP50. The Student *t*-test was conducted to assess the statistical significance (* *p* < 0.05; ** *p* < 0.01; *** *p* < 0.001).

**Figure 6 molecules-28-01826-f006:**
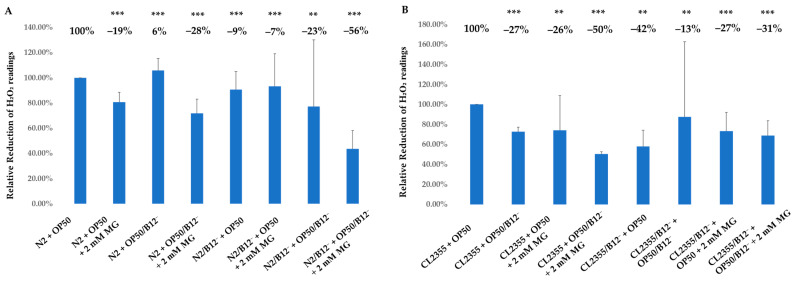
Antioxidant activity of mogrosides in N2, N2/B12^−^, CL2355 and CL22355/B12^−^ worms. (**A**) Synchronized N2 and N2/B12^−^ worms were hatched and grown on either OP50 or OP50/B12^−^ in the presence or absence of 2 mM mogrosides. (**B**) Synchronized CL2355 and CL22355/B12^−^ worms were hatched and grown on either OP50 or OP50/B12^−^ in the presence or absence of 2 mM mogrosides. N2 and CL2355 worms were collected at the L4 stage and assayed for ROS. Three independent experiments were used to calculate the standard deviations. The Student *t*-test was conducted to assess the statistical significance between the treatments and the non-treated N2 (**A**) and CL2355 (**B**) worms (** *p* < 0.01); *** *p* < 0.001.

**Figure 7 molecules-28-01826-f007:**
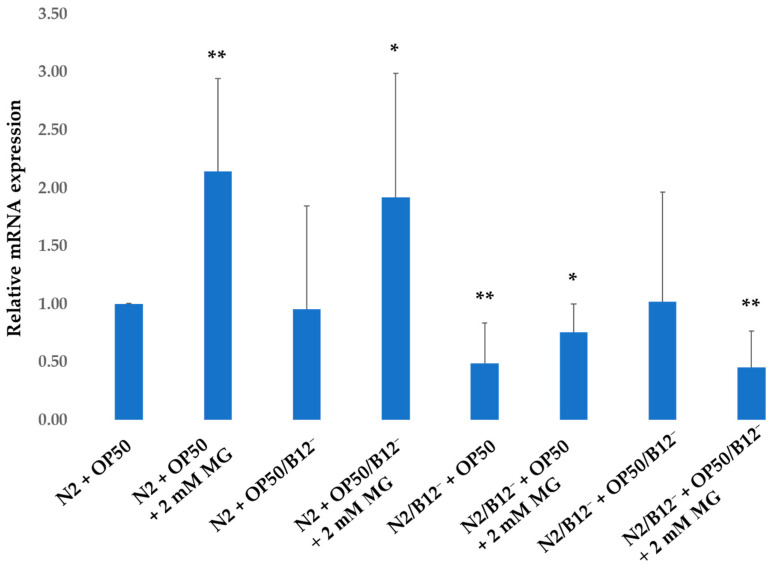
Expression of *lrp-1* gene on N2 and N2/B12^−^ worms fed with either OP50 or OP50/B12^−^ with or without 2 mM mogrosides (MG). The fold-change (2^−ΔΔCt^) and standard deviations of the gene expression were calculated from five independent experiments. All treatments were compared to the N2 + OP50. The Student *t*-test was conducted to assess the statistical significance (* *p* < 0.05; ** *p* < 0.01).

**Figure 8 molecules-28-01826-f008:**
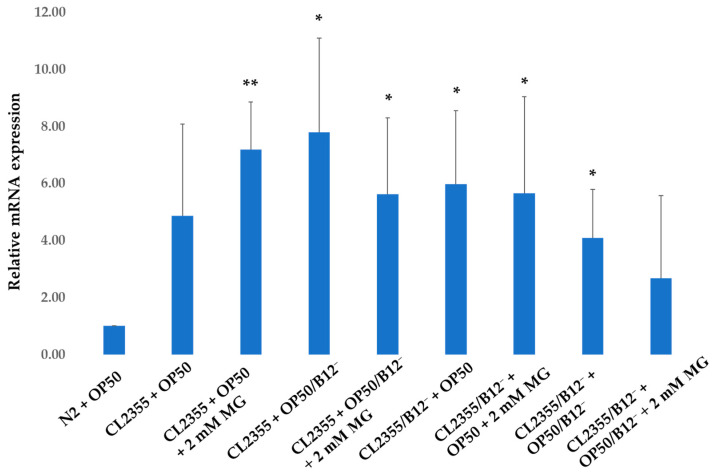
Expression of *lrp-1* gene on CL2355 and CL2355/B12^−^ worms fed with either OP50 or OP50/B12^−^ with or without 2 mM mogrosides (MG). The fold-change (2^−ΔΔCt^) and standard deviations of the gene expression were calculated from three independent experiments. All treatments were compared to N2 + OP50. The Student *t*-test was conducted to assess the statistical significance (* *p* < 0.05; ** *p* < 0.01).

**Figure 9 molecules-28-01826-f009:**
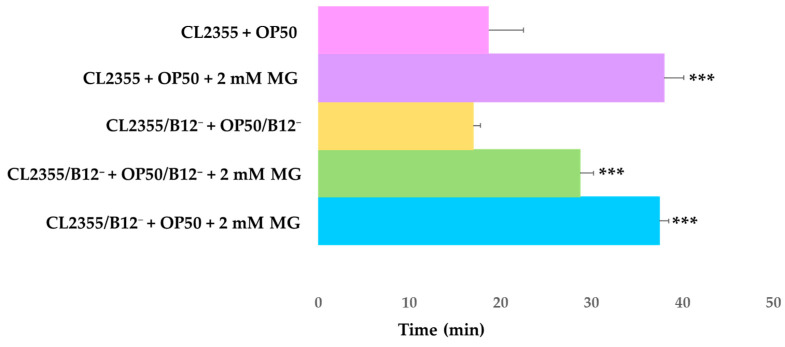
Serotonin sensitivity assay for L4 CL2355 and CL2355/B12^−^ worms treated with mogrosides. Worms were synchronously hatched and raised on NGM plates containing either OP50 or OP50/B12^−^ with or without 2 mM mogrosides at 16 °C for 4–5 days (approximate late L3 stage). Then switched to 22 °C for 24 h until the worms reached the L4 stage. L4 worms were scored for the time it took to complete paralysis. The standard deviations were calculated from four replicates for each treatment. The Student *t*-test was conducted to assess the statistical significance between the treatments and the CL2355 + OP50 or CL2355/B12^−^ + OP50/B12^−^ (*** *p* < 0.001).

**Figure 10 molecules-28-01826-f010:**
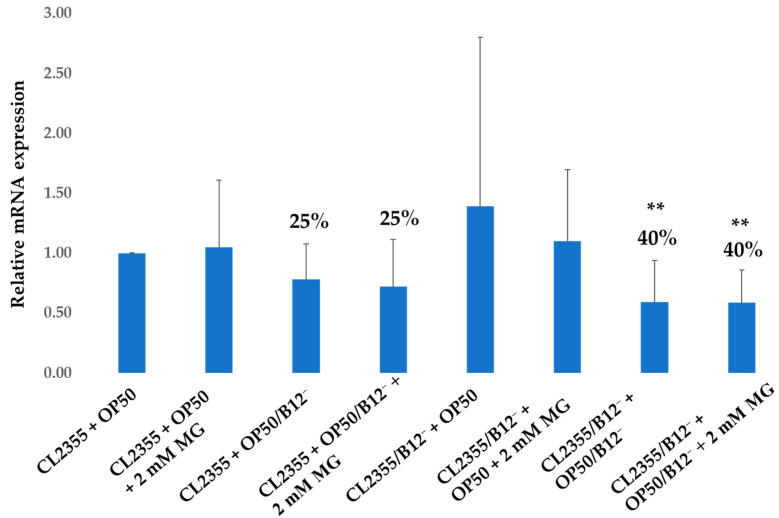
Expression of *ace-2* gene on CL2355 and CL2355/B12^−^ worms fed with either OP50 or OP50/B12^−^ with or without 2 mM mogrosides (MG). The fold-change (2^−ΔΔCt^) and standard deviations of the gene expression were calculated from six independent experiments. All treatments were compared to the control CL2355 fed with only OP50. The Student *t*-test was conducted to assess the statistical significance (** *p* < 0.01).

**Table 1 molecules-28-01826-t001:** LC-MS identification of Mogrosides.

Retention Time (minutes)	Identification	Formula	[M-H]^−^, ppm
5.78	Mogroside IIe	C_42_H_72_O_14_	799.4825, −2.4
3.81	Mogroside III	C_48_H_82_O_19_	961.5364, −0.8
3.42	Mogroside IV	C_54_H_92_O_24_	1123.5894, 0.5
2.81	Mogroside V	C_60_H_102_O_29_	1285.6469, 4.0

**Table 2 molecules-28-01826-t002:** Genes selected for this study. The table lists known functions of the human genes and their *C. elegans* orthologs, along with known biochemical properties of the encoded proteins. Source for *C. elegans* orthologs’ function: https://www.wormbase.org (accessed on 24 December 2022).

Human Genes	*C. elegans* Orthologs	Function
*SOD-1*	*sod-1*	*SOD-1* converts superoxide into less toxic oxygen and hydrogen peroxide. Its mutational inactivation has been linked to amyotrophic lateral sclerosis but its pathological mechanism has not been determined [29]. *sod-1* exhibits superoxide dismutase activity.
*GPX-4*	*gpx-1*	*GPX-4* encodes a glutathione peroxidase, which uses glutathione to catalyze the oxidation of lipids or the reduction of hydrogen peroxide to water [26]. *gpx-1* is known to enable phospholipid hydroperoxide glutathione peroxidase activity.
*CAT*	*ctl-1*	Human catalase (*CAT*) converts hydrogen peroxide to water and oxygen [26]. *ctl-1* encodes a protein with catalase activity.
*GSTP1*	*gst-10*	Human *GSTP1* encodes a member of the glutathione transferases, which conjugate a variety of substrates to glutathione to prevent cellular injury induced by oxidative stress [26]. The *gst-10* gene product exhibits glutathione transferase activity.
*LAMA5*	*epi-1*	*LAMA5* is encodes one of the laminin alpha chains and contributes to the basal membrane of neuromuscular junctions [30]. *epi-1* is predicted to encode a structural component of the extracellular matrix that is involved in the positive regulation of endopeptidase activity and the generation of neurons.
*LRP2*	*lrp-1*	*LRP2* (LDL receptor-related protein 2) participates in the transport and endocytosis of the Aβ complex [31]. *lrp-1* affects sterol transporter activity, regulation of locomotion and larval development. *C. elegans lrp-1* closely resembles *LRP2* and its decrease in expression has been shown to affect neurotransmission in *C. elegans* [18].
*LDL*	*lrx-1*	Low-density lipoprotein (*LDL*) receptor domains are known to play an important part in cholesterol homeostasis. Reduction of *LDL* receptors increases the deposition of Aβ in the brain [32]. *lrx-1* is predicted to encode a protein that has low-density lipoprotein (*LDL*) receptor domains.
*TRAF3*	*trf-1*	*TRAF3* is a tumor necrosis factor gene and is involved in the innate immune response [28]. *trf-1* is predicted to enable both ubiquitin-protein ligase binding and tumor necrosis factor receptor biding activity. It is also involved in innate immune response and the defense response against Gram-negative bacterium.
*TNFA1P1*	*ZC239.12* *F22E5.6*	*TNFA1P1* encodes tumor necrosis factor and is involved in the innate immune response [28]. *ZC239.12* and *F22E5.6* act upstream of, or as part of the IRE1- and PERK-mediated unfolded protein responses.
*HSP*	*hsp-16.2* *hsp-16.41*	*HSP* are heat shock proteins involved in stress response [2]. *hsp-16.2* and *hsp-16.41* gene products are predicted to promote unfolded protein binding activity and be involved in response to heat stress.

**Table 3 molecules-28-01826-t003:** Oligonucleotide primers used in RT-qPCR studies.

Genes	Forward Primer	Reverse Primer	Accession Number
*act-1*	CTCCACGCGCCGTGTT	CATACCGACCATGACTCCTTGA	T04C12.6.1
*sod-1*	GCCGGAGCCCATGGAT	CGGCCTTACAGTACTTGGTGATG	C15F1.7a.1
*gpx-1*	GCGAGGGAGTCGGAGACAA	GAGCTCCGGCGTTTCCA	F26E4.12.1
*ctl-1*	GACCGAATTTGAACGCGTATC	TCGCGTTGATCCAGACTTTGT	Y54G11A.6.1
*gst-10*	ACAAAAAGGATGGTCTCGAAGTTC	TGGTTCCTGACCGGCAAA	Y45G12C.2a.1
*epi-1*	TCACAATTTCCCACCGAAAAC	TCCTGTCAGAGAGCAATAGATTTCA	K08C7.3a.1
*lrp-1*	AGCCGTTCAACCGGTTCTT	ATAAGGCTGCTGCTGAGTTGCT	F29D11.1.1
*lrx-1*	GCGACCCGCTGCAATCTA	GCGGCCAAGAGTGGTGTTAC	T04H1.6.1
*ace-1*	AGCCGTTCAACCGGTTCTT	ATAAGGCTGCTGCTGAGTTGCT	W09B12.1.1
*trf-1*	TGTCAACATGATCGGGCAAA	TCAAAAGTGCAAACGACTGGAA	F45G2.6.1
*F22E5.6*	TCCCCATACGAAACAACACA	CTCCTCCCAGCTTTTCCACAA	F22E5.6.1
*ZC239.12*	CCAGAAGAATCCCCATACGA	TCCTCCTCCAACTTTTCCAAA	ZC239.12.1
*hsp-16.2*	GGTGCAGTTGCTTCGAATCTT	TCTTCCTTGAACCGCTTCTTTC	Y46H3A.3a.1
*hsp-16.41*	AAACAAAATCGGAACATGGATACTT	TGGAGCCTCAATTTGGAGTTTTC	Y46H3A.2.1

## Data Availability

Not applicable.

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
