# Peer review of "Potential Anti-Alzheimer Properties of Mogrosides in Vitamin B12-Deficient Caenorhabditis elegans"

_molecules, 2023, doi:10.3390/molecules28041826_

Round 1

Reviewer 1 Report

The authors studied the effect of the mogrosides glycosides on B12-deffecient wild type and transgenic type of Caenorhabditis elegans worms. Comments are below.

Key words: please write the worm’s name in full to avoid misinterpretation with other same abbreviated name of different organisms.

Introduction: please, do not start with the aim of the study. First discuss the background, then the gap for the study’s sake and lastly the aim. Please provide the chemical structures for the most prevalent mogrosides in the extract under investigation.

Results: the data for the chemical analysis characterization must be provided, is there any reason for not providing these data as you already did the analyses as claimed in the manuscript.

It is advisable to provide photos for the worms and eggs before and after treatment with mogrosides extract in a good resolution.

Discussion: try to relate or expect the chemical motif in mogrosides structure responsible for the effects shown.

Material and methods:

1-      Discuss in more detail the extraction process and the characterization verification analyses were not mentioned at all in the methods section, please add.

2-      Add the accession numbers for the studied genes.

References: The Latin names for organisms and plants must be italic, please revise.

Author Response

Reviewer #1:

The authors studied the effect of the mogrosides glycosides on B12-deffecient wild type and transgenic type of Caenorhabditis elegans worms. Comments are below.

Key words: please write the worm’s name in full to avoid misinterpretation with other same abbreviated name of different organisms.

Page 1/Line 30: Full worm’s name has been added.

Introduction: please, do not start with the aim of the study. First discuss the background, then the gap for the study’s sake and lastly the aim.

Page 1-2/Line 33-154: We have revised the introduction.

Please provide the chemical structures for the most prevalent mogrosides in the extract under investigation.

Page 2-3/Line 156-245: We extensively revised our Results in 2.1. Chemical Composition for mogrosides.  Our mogroside extract was a gift received a long ago from Ibis America.  We added the citation for its extraction and purification.  Since it was impossible to retrieve the original HPLC data, we recently re-analyzed it by the UPLC-qToF-MS method and provided the data in Figure 1 and Table 1.  The structures and molecular weights of mogroside IIe, III, IV and V were obtained from https://pubchem.ncbi.nlm.nih.gov and provided in Supplementary Figure S1 and Table S1.  These data show that mogroside V, known to be the active component of mogroside mixture, was the most abundant in our extract.

Results: the data for the chemical analysis characterization must be provided, is there any reason for not providing these data as you already did the analyses as claimed in the manuscript.

Page 2-3/Line 156-245: Section 2.1. Chemical Composition has been revised. See above.

It is advisable to provide photos for the worms and eggs before and after treatment with mogrosides extract in a good resolution.

Pictures of the worms and eggs were not taken during the experiment process. We will take this into consideration for future experiments.

Discussion: try to relate or expect the chemical motif in mogrosides structure responsible for the effects shown.

Page 13/ Line 545-548: We have added more description and discussion related to the composition of our extracted mogrosides and the relation to the effects shown in our experiments.

Material and methods:

1-      Discuss in more detail the extraction process and the characterization verification analyses were not mentioned at all in the methods section, please add.

Page 14/Lines 654-675: The extraction process and the characterization verification analysis have been extensively revised.  Also see above.

2-      Add the accession numbers for the studied genes.

References: The Latin names for organisms and plants must be italic, please revise.

Page 17-19: The references have been revised.

Reviewer 2 Report

The paper entitled “Potential anti-Alzheimer properties of mogrosides in vitamin B12-deficient C. elegans“ by Shi et al.  investigated the potential anti-AD effect of mogrosides in vitamin B12-deficient wild-type N2 and in transgenic CL2355 C. elegans expressing amyloid-β peptide. The data indicated that  mogrosides may exert preventative rather than curative effects that counteract the detrimental vitamin B12-deficient environment in N2 and CL2355 C. elegans by modulating oxidation-related gene expression.The research is interesting.  The overall level of the paper is well written. This paper has a potential to be accepted. Some minor suggestions were as follows:

Line 84-86: the UV and HPLC spectrum of mogroside should be included in the supplementary data.

Line 66: There should be a "." after [21].

Author Response

Reviewer #2:

The paper entitled “Potential anti-Alzheimer properties of mogrosides in vitamin B12-deficient C. elegans“ by Shi et al.  investigated the potential anti-AD effect of mogrosides in vitamin B12-deficient wild-type N2 and in transgenic CL2355 C. elegans expressing amyloid-β peptide. The data indicated that  mogrosides may exert preventative rather than curative effects that counteract the detrimental vitamin B12-deficient environment in N2 and CL2355 C. elegans by modulating oxidation-related gene expression. The research is interesting.  The overall level of the paper is well written. This paper has a potential to be accepted. Some minor suggestions were as follows:

Line 84-86: the UV and HPLC spectrum of mogrosides should be included in the supplementary data.

Page 2-3/Line 156-245: UPLC-qToF-MS analysis have been added to section 2.1. Chemical Composition.  Structures and molecular weights of mogrosides have been added as Supplementary Figure S1 and Table S1. https://pubchem.ncbi.nlm.nih.gov

Line 66: There should be a "." after [21].

Page 2/Line 142: “.” has been added in the after [21->23]. (Changed after revision of Introduction)

Reviewer 3 Report

Remarks to the Author:

Denia Cai Shi and colleagues explored potential anti-alzheimer properties of mogrosides in vitamin b12-deficient C. elegans. They discovered that mogrosides have a beneficial effect on the lifespan and egg-laying rate of N2 and vitamin B12-deficient N2 worms, can effectively delay the paralysis of CL2355 worms in the serotonin sensitive assay. and increase the expression of oxidative protective genes in N2 worms fed with vitamin B12-deficient OP50 bacterium.

It is an interesting work research, but there are many pitfalls and caveats from the presented results.

Overall, I think the manuscript should be reconsidered after major revision.

Major points:

1. The logic of introduction needs to be reorganized. The introduction should include the research background, research purpose and significance, and should not involve the results. Such as “Our results indicate…” in line 76.

2. The authors explored the expression level of related genes. Why didn't you study the change of protein level?

Minor points:

3. The full name should be used when it appears in abstract and introduction for the first time, such as Alzheimer’s disease (line 28) and Caenorhabditis elegans (line 16).

4. It would be better to add a short conclusion after each result.

5. line 134: This paragraph is summarized by the author with reference to other literature, not the experimental results, and can be combined with the next paragraph.

6. Why use mogrosides with a concentration of 2 mM? Have you done any experiments on concentration gradient or what is the reference basis??

7. The format of references in the text should be uniform, and the references in line 381 should be represented by number.

Author Response

Reviewer #3:

Remarks to the Author:

Denia Cai Shi and colleagues explored potential anti-alzheimer properties of mogrosides in vitamin b12-deficient C. elegans. They discovered that mogrosides have a beneficial effect on the lifespan and egg-laying rate of N2 and vitamin B12-deficient N2 worms, can effectively delay the paralysis of CL2355 worms in the serotonin sensitive assay. and increase the expression of oxidative protective genes in N2 worms fed with vitamin B12-deficient OP50 bacterium.

It is an interesting work research, but there are many pitfalls and caveats from the presented results.

Overall, I think the manuscript should be reconsidered after major revision.

Major points:

  1. The logic of introduction needs to be reorganized. The introduction should include the research background, research purpose and significance, and should not involve the results. Such as “Our results indicate…” in line 76.

Page 1-2/Line 33-154: Logic of the introduction has been revised and results been deleted from the introduction.

  1. The authors explored the expression level of related genes. Why didn't you study the change of protein level?

Gene expression level changes were analyzed by RT-qPCR in this study, which is the common method nowadays.  Since we analyzed many genes in this study, we could not find antibodies against these gene products.  Additionally, most of the mRNA levels were up-regulated in the lower range, it would be difficult to see the changes at the protein levels.  We would definite consider adding this approach in our future studies.

Minor points:

  1. The full name should be used when it appears in abstract and introduction for the first time, such as Alzheimer’s disease (line 28) and Caenorhabditis elegans(line 16).

Page 1/Line 30: Caenorhabditis elegans has been corrected.

Page 1/Line 33: “AD” has been changed to “Alzheimer’s disease (AD)”. Introduction has been revised.

  1. It would be better to add a short conclusion after each result.

A short conclusion has been added to each result section.

  1. line 134: This paragraph is summarized by the author with reference to other literature, not the experimental results, and can be combined with the next paragraph.

Revised.

  1. Why use mogrosides with a concentration of 2 mM? Have you done any experiments on concentration gradient or what is the reference basis??

Our preliminary data on lifespan analysis with 5 mM, 4 mM, 3 mM and 2 mM of mogrosides showed that higher than 2 mM concentration slightly decreased the lifespan of the C. elegans, so we chose 2 mM concentration throughout this study.

  1. The format of references in the text should be uniform, and the references in line 381 should be represented by number.

Page 14: Reference is represented by number in line 656.  Revised.

Round 2

Reviewer 1 Report

The authors have made the corrections. Only, The accession numbers of the studied genes are required to be added.

Author Response

Dear editor:

Thanks to the reviewers’ comments for our revised manuscript!  There is only one comment from Reviewer #1 that need to address.  Please see below.

Reviewer #1 comment:

The authors have made the corrections. Only, The accession numbers of the studied genes are required to be added.

Our response:

We have added gene accession numbers for all the genes studied in Table 3.

The revised manuscript in tracking function is attached 

Thank you again for your help!

Ron Di et al. 2/10/2023

Reviewer 3 Report

The author and colleagues have answered and revised all the questions raised. Therefore, I support the publication of this study.

Author Response

Thank you for agreeing to publish our paper!